# SFi-Former: Sparse Flow induced Attention for Graph Transformer

## Abstract

Graph Transformers (GTs) have demonstrated superior performance compared to traditional message-passing graph neural networks in many studies, especially in processing graph data with long-range dependencies. However, GTs tend to suffer from weak inductive bias, overfitting and over-globalizing problems due to the dense attention. In this paper, we introduce SFi-attention, a novel attention mechanism designed to learn sparse pattern by minimizing an energy function based on network flows with $\ell_1$-norm regularization, to relieve those issues caused by dense attention. Furthermore, SFi-Former is accordingly devised which can leverage the sparse attention pattern of SFi-attention to generate sparse network flows beyond adjacency matrix of graph data. Specifically, SFi-Former aggregates features selectively from other nodes through flexible adaptation of the sparse attention, leading to a more robust model. We validate our SFi-Former on various graph datasets, especially those graph data exhibiting long-range dependencies. Experimental results show that our SFi-Former obtains competitive performance on GNN Benchmark datasets and SOTA performance on Long-Range Graph Benchmark (LRGB) datasets. Additionally, our model gives rise to smaller generalization gaps, which indicates that it is less prone to over-fitting.

## 1 Introduction

Traditional graph representation learning methods, such as GCN (Defferrard et al., 2016; Kipf & Welling, 2016; Zhang et al., 2019), GAT (Veličković et al., 2017), GIN (Xu et al., 2018) and GatedGCN (Bresson & Laurent, 2017), typically rely on a local message-passing mechanism that integrates the features of a node's neighbors with those of directly or closely connected nodes. This design effectively captures the topological structure of the graph, but it faces issues such as over-smoothing (Oono & Suzuki, 2019), over-squashing (Alon & Yahav, 2020), and an inability to handle graph data with long-range dependencies. As the transformer architectures have achieved widespread successes in other domains, it also receives a growing interests to graph learning. To this end, Graph Transformers (GTs) have been proposed, enabling each node to interact with all other nodes in the graph through self-attention mechanism (Dwivedi & Bresson, 2021; Ying et al., 2021; Müller et al., 2024). Such short-cut connections between nodes are in sharp contrast with message-passing based GNNs, making GTs beneficial for many realistic applications such as generating molecular graphs (Mitton et al., 2021), generating texts from knowledge graphs (Koncel-Kedziorski et al., 2019), improving recommendation systems (Li et al., 2023) and so on. However, GTs effectively operates on an auxiliary fully-connected graph, disregarding the original graph structure of the problem, which results in a weak inductive bias with respect to the graph's topology (Wang et al., 2024). To address this weakness, positional and structural encodings (PE/SE), such as graph Laplacian eigenvectors (Makarov et al., 2021; Kreuzer et al., 2021a), are commonly used in GTs to incorporate structural information from the original graph. To combine the strengths of message-passing GNNs and GTs, GraphGPS framework is proposed, providing a flexible platform for experimenting with new model designs and learning methods (Rampášek et al., 2022).

Recent works (Shirzad et al., 2023; Fournier et al., 2023) have shown the effectiveness of using sparsity for simplifying the computational complexity of GTs. In this study, we aim to explore another potential advantage of sparsity in improving the performance and stability of GTs, through the design of novel sparse attention mechanisms. In vanilla transformers, each node aggregates features from all other nodes, making it susceptible to attending to irrelevant or spurious information,

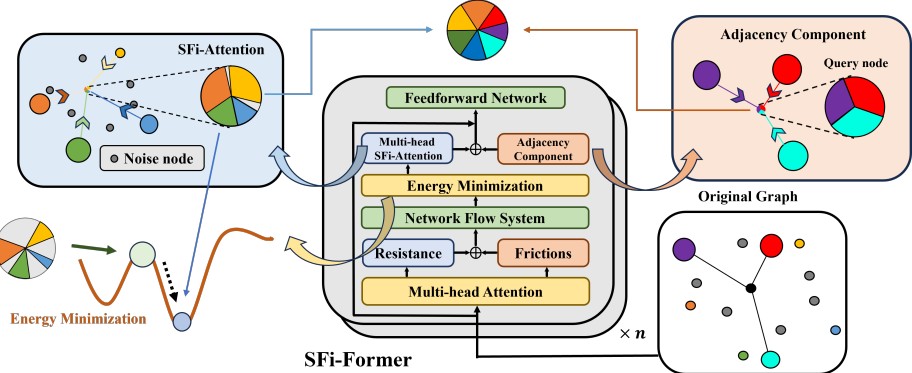

Figure 1: Overview of the SFi-Former architecture. Our design enables node features to be aggregated from the features of their adjacent nodes and selectively from distant nodes based on our Sparse-Flow-induced attention mechanism, achieving robust performance on downstream tasks.

which is particularly harmful for problems of small and medium sizes. This has been shown in prior studies, such as the large gaps between the training and testing evaluation metrics in dense GTs (Dwivedi et al., 2022b), as well as in our own experiments presented below. According to the conventional wisdom in statistics, sparse variable selection and other shrinkage methods are helpful in reducing variance of estimates and improving generalization (Hastie et al., 2009). Therefore, we aim to develop adaptive sparse transformers that are better suited for graph-related tasks where each node selectively aggregates information from other nodes, in order to enhance performance and improve generalization. We do not aim to address the computational bottleneck of GTs at this point.

To achieve this, we draw inspiration from recent advancements in a study of flow-based semi-supervised learning (Rustamov & Klosowski, 2018). In this approach, an unlabeled node of interest is treated as a sink node that receives sparse network flows from a set of labeled nodes. The flow patterns are determined by minimizing an energy function, and the unlabeled node gathers information from each labeled node, with the weight of the information based on the total outflow from each labeled node. For our purposes, we treat each query node as a sink node that receives sparse flows from key nodes, with the magnitudes of these flows determining the attention scores. Further motivated by recent findings that GTs tend to overly focus on distant nodes (Xing et al., 2024), we enhance the sparse-flow-based attention with hard-wired local connections with adjacent nodes. This allows the sparse-flow attention to focus on the residual information in addition to those from adjacency components, which further exploits the graph structure information. We refer to our GT architecture based on the Sparse-Flow-induced attention mechanism as SFi-Former, where the architecture is shown in Figure 1.

Our main **contributions** are as follows. (1) We propose SFi-Former, an adaptive sparse attention mechanism induced by sparse network flows from key nodes to query nodes. It demonstrates robust performance on capturing dependencies among nodes. (2) We propose an energy-based framework for attention, which incorporates the standard self-attention mechanism as a special case, and provides a flexible framework to accommodate additional modeling elements. (3) Built upon the recent GraphGPS framework (Rampasek et al., 2022), our SFi-Former outperform alternatives in processing graph data with long-range dependencies, which achieves SOTA performance on the LRGB datasets. It can alleviate overfitting compared to GTs with dense attention mechanism and demonstrates competitive performance across various graph datasets.

## 2 RELATED WORK

**Graph Transformers (GTs).** Recently, transformer architectures and attention mechanisms (Vaswani et al., 2017) have achieved tremendous successes in natural language processing (NLP) (Kalyan et al., 2021) and computer vision (CV) (d'Ascoli et al., 2021; Guo et al., 2021; Han et al., 2022), with growing efforts to apply them to graph structures as well. However, because graph transformers (GTs) rely on global attention mechanisms, they suffer from a weak inductive

bias, limiting their ability to fully exploit graph structure. Later research introduced various positional encoding (PE) methods, such as SAN (Kreuzer et al., 2021b), Graphormer (Ying et al., 2021), and SAT (Chen et al., 2022a), within the transformer framework. Concurrently, structural encoding (SE) methods (Dwivedi et al., 2022a; Bodnar et al., 2021; Bouritsas et al., 2022) were also developed. Both PEs and SEs aim to mitigate the weak inductive bias problem. GraphGPS (Rampasek et al., 2022) offers a unified framework that integrates positional and structural encodings in GTs. Nevertheless, both PE and SE may still be insufficient to fully capture the inductive bias of the graph structure.

**Sparse transformers.** The original transformer architecture (Vaswani et al., 2017) has the quadratic complexity in the number of tokens, which becomes a bottleneck for processing long sequences in NLP tasks. Various sparse transformers, including Performer (Choromanski et al., 2021), Big-Bird (Zaheer et al., 2020), and Reformer (Kitaev et al., 2020), have been developed to address this bottleneck (Catania et al., 2023). However, these methods have not demonstrated competitive performance on graph data with long-range dependencies (Rampasek et al., 2022). Sparse attention has also been considered in message-passing based Graph Attention Networks (Ye & Ji, 2021). Exphormer leverages the idea of virtual global nodes and expander graphs to create sparse GTs (Shirzad et al., 2023). While such sparse transformers effectively reduce computation, they are often sub-optimal for enhancing performance.

**Energy-based graph models.** In addition to traditional GNNs and GTs, graph neural diffusion models and energy-based graph models are significant research areas for learning from graph data (Chamberlain et al., 2021; Bronstein et al., 2021). The work by (Rustamov & Klosowski, 2018) introduces a flow-based model for semi-supervised learning, improving label propagation. Elastic Graph Neural Networks (EGNN) (Liu et al., 2021) employ $\ell_1$ and $\ell_2$-minimizaiton induced graph smoothing for semi-supervised learning. Graph Implicit Nonlinear Diffusion (GIND) (Chen et al., 2022c) proposes a method for feature aggregation using non-linear diffusion induced by the optimization of an energy function. Additionally, DIGNN (Fu et al., 2023) introduced implicit GNN layers as fixed-point solutions to Dirichlet energy minimization. The survey by (Han et al., 2023) provides a good overview of this growing area. However, these studies do not address GTs, which are the main focus of the current work.

## 3 FLOW INDUCED ATTENTION PATTERNS

In this section, we outline the derivations of attention patterns based on energy-based flow network, aiming to extend the existing Transformer architecture and develop a flexible framework that can learn the optimal sparsity dynamically.

### 3.1 ELECTRIC CIRCUIT VIEW OF SELF-ATTENTION

The standard self-attention mechanism with $n$ tokens can be represented as interactions on a bi-directional fully-connected graph $\mathcal{G}(\mathcal{V}, \mathcal{E})$ with $n = |\mathcal{V}|$ nodes (Vaswani et al., 2017). Denoting the feature vectors of these $n$ tokens as $\boldsymbol{X} \in \mathbb{R}^{n \times d}$, the attention for the $h$-th head can be expressed as $\mathrm{ATT}^h(\boldsymbol{X}) = \mathrm{Softmax}\left(\frac{(\boldsymbol{X}\boldsymbol{W}_K^h)(\boldsymbol{X}\boldsymbol{W}_Q^h)^T}{\sqrt{d_k}}\right)^1$ where $\boldsymbol{W}_K^h$ and $\boldsymbol{W}_Q^h \in \mathbb{R}^{d \times d_k}$. The forward step of the standard attention mechanism at the $k$-th layer is defined as follows:

$$\boldsymbol{X}_i^{(k+1)} = \boldsymbol{X}_i^{(k)} + \sum_{h=1}^H \sum_{j=1}^n \mathrm{ATT}^h(\boldsymbol{X}^{(k)})_{i,j} \boldsymbol{X}_j^{(k)} \boldsymbol{W}_V^h \boldsymbol{W}_O^h, \tag{1}$$

where $\boldsymbol{W}_V^h \in \mathbb{R}^{d \times d_V}, \boldsymbol{W}_O^h \in \mathbb{R}^{d_V \times d}$ and a residual connection has been introduced. The forward step in Eq. (1) indicates that the larger $\mathrm{ATT}^h(\boldsymbol{X}^{(k)})_{i,j}$ is, the greater the contribution of the $j$-th token's feature $\boldsymbol{X}_j^{(k)}$ to the $i$-th token's feature $\boldsymbol{X}_i^{(k+1)}$ at the next layer.

Here, we re-interpret self-attention through the lens of electric circuits on fully-connected graphs. Let the node set be $\mathcal{V} = \{\boldsymbol{v}_1, \boldsymbol{v}_2, \cdots, \boldsymbol{v}_n\}$. Consider a query node $\boldsymbol{v}_s \in \mathcal{V}$, where each node $\boldsymbol{v}_i$ (including node $\boldsymbol{v}_s$) has a short-cut link to node $\boldsymbol{v}_s$. Let node $\boldsymbol{v}_s$ act as a sink, which will draw one unit of

---

[1]The softmax operator on a matrix $\boldsymbol{X} \in \mathbb{R}^{n \times n}$ is defined as $\mathrm{Softmax}(\boldsymbol{X}) = \frac{\exp(\boldsymbol{X})}{\exp(\boldsymbol{X})\mathbf{1}_n \mathbf{1}_n^T}$, where $\exp(\cdot)$ denotes an elementwise exponential. Componentwise, this can be expressed as $[\mathrm{Softmax}(\boldsymbol{X})]_{ij} = \frac{\exp(X_{ij})}{\sum_k \exp(X_{ik})}$.

resources from all other nodes in the graph through the short-cut links. Each node $\boldsymbol{v}_i$ has to transport some amount of resources to the sink node $\boldsymbol{v}_s$ to satisfy its demand. Let $z_i$ represent the network flow from node $\boldsymbol{v}_i$ to node $\boldsymbol{v}_s$, they satisfy the flow conservation constraint $\sum_{i=1}^n z_i = 1$. The amount of network flow $z_i$ is dictated by a distance measure $r_i$ between node $\boldsymbol{v}_i$ and node $\boldsymbol{v}^*$; we refer to $r_i$ as the resistance on the $i$-th link. Following the Thomson's Principle for resistor networks (Doyle & Snell, 1984), the optimal network flows are obtained by solving a quadratic energy minimization problem along with its corresponding Lagrangian function as

$$\min_{\boldsymbol{z}} E(\boldsymbol{z}) = \frac{1}{2} \boldsymbol{z}^T \boldsymbol{R} \boldsymbol{z} \quad \text{s.t.} \quad \boldsymbol{z}^T \mathbf{1}_n - 1 = 0, \tag{2}$$

$$\mathcal{L}(\boldsymbol{z}, \mu) = \frac{1}{2} \boldsymbol{z}^T \boldsymbol{R} \boldsymbol{z} - \mu(\boldsymbol{z}^T \mathbf{1}_n - 1), \tag{3}$$

where $\boldsymbol{z} = (z_1, \cdots, z_n)^T$ and $\boldsymbol{R} = \mathrm{diag}(r_1, \cdots, r_n)$. The lagrange multiplier $\mu$ can be interpreted as the negative electric potential of node $\boldsymbol{v}_s$ and the electric potentials at other nodes are zero since their outflows $\boldsymbol{z}$ are unconstrained, which effectively have zero Lagrange multipliers (Rebeschini & Tatikonda, 2019). By solving $\frac{\partial \mathcal{L}}{\partial \boldsymbol{z}} = 0$, we can obtain the optimal flow as $z_i^* = \mu/r_i = [0 - (-\mu)]/r_i$, which satisfies the Ohm's Law $I = U/R$. Furthermore, by solving $\frac{\partial \mathcal{L}}{\partial \mu} = 0$, we can obtain the negative electric potential at node $\boldsymbol{v}_s$ as $\mu^* = 1/\sum_{i=1}^n (r_i)^{-1}$. Note that $1/\sum_{i=1}^n (r_i)^{-1}$ corresponds to the total resistance of resistors $\{r_i\}$ connected in parallel, and the optimal network flow at the $i$-th link is $z_i^* = \frac{(r_i)^{-1}}{\sum_{j=1}^n (r_j)^{-1}}$. If we identify the resistance as $r_i \propto \exp(-\boldsymbol{q}_s^T \boldsymbol{k}_i/\sqrt{d_k})^2$, we observe that the optimal network flow is $z_i^* = \frac{\exp(\boldsymbol{q}_s^T \boldsymbol{k}_i/\sqrt{d_k})}{\sum_{j=1}^n \exp(\boldsymbol{q}_s^T \boldsymbol{k}_j/\sqrt{d_k})}$, which is the attention score from the query node $\boldsymbol{v}_s$ to a key node $\boldsymbol{v}_i$ in the standard self-attention mechanism. Intuitively, the greater the alignment between the query vector $\boldsymbol{q}_s$ and the key vector $\boldsymbol{k}_i$, the smaller the resistance $r_i$, resulting in a higher electrical flow $z_i^*$ from node $\boldsymbol{v}_i$ to $\boldsymbol{v}_s$, which implies a greater attention from node $\boldsymbol{v}_s$ to $\boldsymbol{v}_i$.

To enumerate different query nodes, it is convenient to express the above formalism in matrix notation. Let $\boldsymbol{Z} \in \mathbb{R}^{n \times n}$ denote the flow pattern with $Z_{i,j}$ being the flow from node $\boldsymbol{v}_j$ to the sink node $\boldsymbol{v}_i$, and $\boldsymbol{R}^h \in \mathbb{R}^{n \times n}$ denotes the resistances on the corresponding links. The energy minimization problem of Eq. (2) can then be written as

$$\min_{\boldsymbol{Z}} E^h(\boldsymbol{Z}) = \frac{1}{2} \mathrm{Tr}\left[ (\boldsymbol{R}^h \circ \boldsymbol{Z}) \boldsymbol{Z}^T \right] \quad \text{s.t.} \quad \boldsymbol{Z}\mathbf{1}_n - \mathbf{1}_n = \mathbf{0}_n. \tag{4}$$

where $\circ$ represents the Hadamard product. The corresponding optimal flows are given by

$$Z_{i,j}^{*h} = \frac{(R_{i,j}^h)^{-1}}{\sum_{k=1}^n (R_{i,k}^h)^{-1}}. \tag{5}$$

If we choose the trainable resistances as $\boldsymbol{R}^h = \mathrm{Softmax}\left( -\frac{(\boldsymbol{X}\boldsymbol{W}_K^h)(\boldsymbol{X}\boldsymbol{W}_Q^h)^T}{\sqrt{d_k}} \right)$, we obtain the optimal flows as $\boldsymbol{Z}^{*h} = \mathrm{Softmax}\left( \frac{(\boldsymbol{X}\boldsymbol{W}_K^h)(\boldsymbol{X}\boldsymbol{W}_Q^h)^T}{\sqrt{d_k}} \right) = \mathrm{ATT}^h(\boldsymbol{X})$. Therefore, optimizing the energy function in Eq. (4) recovers the conventional self-attention pattern $\mathrm{ATT}^h(\boldsymbol{X})$. This perspective provides a framework for designing other attention mechanisms by adjusting the energy function.

### 3.2 Sparse-Flow-induced Attention (SFi-Attention)

As outlined in Sec. 1, our objective is to devise sparse transformers by modifying the quadratic energy function of network flows. To this end, we introduce an additional $\ell_1$-norm penalty to the network flow in Eq. (2) to encourage sparsity in the flow patterns. Minimization based on the $\ell_1$-norm is widely employed across various fields, with prominent examples including LASSO in statistics (Hastie et al., 2009) and compressed sensing (Wright & Ma, 2022). In LASSO for regression problems, the $\ell_1$-norm regularization performs shrinkage to the regression coefficients, which can significantly reduce the estimation variance for high dimensional problems; it is also able to shrink coefficients to zero, producing sparse solutions and effectively performing variable selection.

For our purpose, we consider a node $\boldsymbol{v}_s$ as the sink node which draws one unit of resources from all nodes as before. The energy minimization problem for sparse network flows and the corresponding

---

[2]Here, $\boldsymbol{q}_s = \boldsymbol{X}_s \boldsymbol{W}_Q$ is the query vector of node $\boldsymbol{v}_s$, and $\boldsymbol{k}_i = \boldsymbol{X}_i \boldsymbol{W}_K$ is the key vector of node $\boldsymbol{v}_i$.

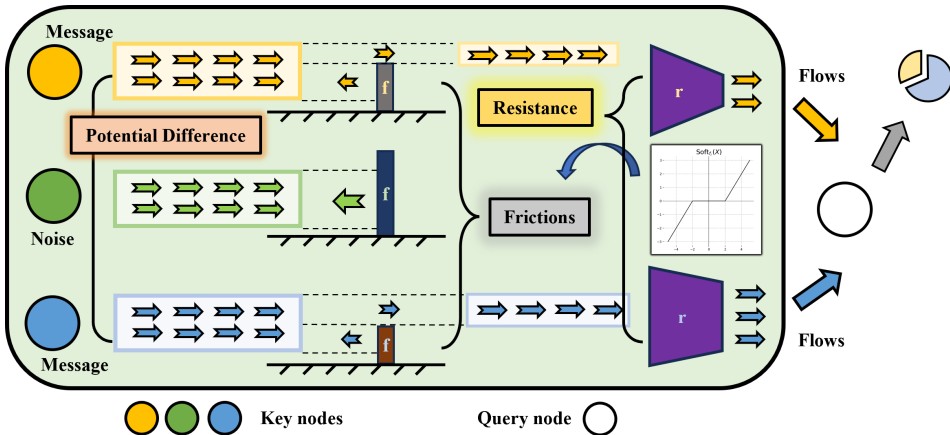

Figure 2: Illustration of the SFi-attention, where the energy-minimized flow $z_i^* = \frac{\text{Soft}_{\lambda f_i}(\mu^*)}{r_i}$ serves as the attention score from a given query node to the key node $\boldsymbol{v}_i$. Here, frictions serve as learnable node-wise noise filters, allowing only strong signals to pass through. The resistance $r_i$ represents a dissimilarity measure of the query vector and the key vector of node $\boldsymbol{v}_i$. The optimal flows $\{z_i^*\}$ correspond to the attention pattern from the given query node to all key nodes.

Lagrangian function are given as follows:

$$\min_{\boldsymbol{z}} E(\boldsymbol{z}) = \frac{1}{2}\boldsymbol{z}^T\boldsymbol{R}\boldsymbol{z} + \lambda||\boldsymbol{f} \circ \boldsymbol{z}||_1 \quad \text{s.t.} \quad \boldsymbol{z}^T\mathbf{1}_n - 1 = 0, \tag{6}$$

$$\mathcal{L}(\boldsymbol{z}, \mu) = \frac{1}{2}\boldsymbol{z}^T\boldsymbol{R}\boldsymbol{z} + \lambda||\boldsymbol{f} \circ \boldsymbol{z}||_1 - \mu(\boldsymbol{z}^T\mathbf{1}_n - 1), \tag{7}$$

where $\lambda$ is a trade-off parameter balancing the $\ell_1$ penalty and the quadratic energy, and the parameters $\boldsymbol{f} = (f_1, \cdots, f_n)^T$ act as element-wise frictions. For convenience, we introduce the soft-thresholding operator $\text{Soft}_\tau(\omega) = \text{sgn}(\omega)\max(|\omega| - \tau, 0)$, where $\tau > 0$. This operator filters out any input signal $\omega$ with a magnitude below $\tau$ (i.e., $|\omega| < \tau$) and shrinks $\omega$ toward zero when $|\omega| \geq \tau$.

The optimality condition $0 \in \frac{\partial\mathcal{L}}{\partial\boldsymbol{z}}$ gives rise to the optimal flow as $z_i^* = \frac{\text{Soft}_{\lambda f_i}(\mu)}{r_i}$, where the Lagrangian multiplier $\mu$ satisfies $\sum_{i=1}^n z_i^* = \sum_{i=1}^n \frac{\text{Soft}_{\lambda f_i}(\mu)}{r_i} = 1$. Physically, the $i$-th link admits a non-zero network flow only if the potential difference $\mu$ between node $\boldsymbol{v}_s$ and node $\boldsymbol{v}_i$ exceeds the friction $f_i$, resulting in sparse flow patterns. The parameter $r_i$ acts as a resistance, relating the shrunk potential difference $\text{Soft}_{\lambda f_i}(\mu)$ to the flow $z_i^*$, similar to the behavior in electric circuits discussed in Sec. 3.1. The key difference is that the sparse flow network exhibits non-linear current-voltage characteristics. Such non-linear circuits were employed in (Rustamov & Klosowski, 2018) to tackle semi-supervised learning problems, which used hand-crafted resistance parameters.

For our purpose, we leverage the framework of sparse network flow optimization to develop sparse attention mechanisms, where the resistances depend on trainable parameters, having the form of $r_i \propto \exp(-\boldsymbol{q}_s^T\boldsymbol{k}_i/\sqrt{d_k})$. Additionally, we notice that the sparsity of flow patterns is influenced by (i) the global balancing parameter $\lambda$, and (ii) the node-specific friction parameters $\{f_i\}$. In light of this, we also make the frictions $\boldsymbol{f}$ depend on trainable parameters, serving as node-wise noise filters to eliminate small flows (attention scores). This design provides a more flexible attention mechanism compared to standard self-attention. Figure 2 clearly illustrates the roles of resistance and friction in the our sparse attention mechanism.

Finally, we note that optimization-induced sparse attention mechanisms have been explored in NLP tasks, such as in the work by (Correia et al., 2019), which achieves this by optimizing an objective function based on Tsallis entropies. Our approach, inspired by network flow problems, differs from these studies and offers a more flexible modeling framework including learnable friction terms as noise filters. Moreover, the network flow problem does not have to be defined on the complete graph as assumed here and other energy functions can also be explored. Extending the flow-based

framework to more general graph topologies and other objective function are interesting directions for future research.

## 3.3 SPECIFYING AND COMPUTING SFI-ATTENTION

We express the sparse flow problem in matrix notation to consider different query nodes,

$$\min_{\boldsymbol{Z}} E_S^h(\boldsymbol{Z}) = \frac{1}{2} \operatorname{Tr}\left[(\boldsymbol{R}^h \circ \boldsymbol{Z})\boldsymbol{Z}^T\right] + \lambda ||\boldsymbol{F}^h \circ \boldsymbol{Z}||_{1,1} \quad \text{s.t.} \quad \boldsymbol{Z}\mathbf{1}_n - \mathbf{1}_n = \mathbf{0}_n, \tag{8}$$

where $\boldsymbol{Z} \in \mathbb{R}^{n \times n}$ is the flow pattern with $Z_{i,j}$ being the flow from node $\boldsymbol{v}_j$ to the sink node $\boldsymbol{v}_i$, and $\boldsymbol{R}^h$ and $\boldsymbol{F}^h$ are the corresponding resistance and friction parameters, respectively. In Eq. (8), $||\boldsymbol{A}||_{1,1}$ represents the entry-wise $L_{1,1}$-norm of the matrix $\boldsymbol{A}$, defined as $||\boldsymbol{A}||_{1,1} = \sum_{i,j} |A_{i,j}|$.

Building on the rationale in Sec. 3.1, we parameterize the resistances as $\boldsymbol{R}^h = \operatorname{Softmax}\big(-\frac{(\boldsymbol{X}\boldsymbol{W}_K^h)(\boldsymbol{X}\boldsymbol{W}_Q^h)^T}{\sqrt{d_k}}\big)$, with trainable parameters $\boldsymbol{W}_K^h$ and $\boldsymbol{W}_Q^h$, which essentially corresponds to conventional multi-head attention. For the friction parameters $\boldsymbol{F}^h$, there are various possible choices. In this work, we primarily define $\boldsymbol{F}^h$ as another multi-head attention with a different set of trainable parameters, $\tilde{\boldsymbol{W}}_K^h$ and $\tilde{\boldsymbol{W}}_Q^h$, though other parameterizations for $\boldsymbol{F}^h$ are also possible.

Due to the non-differentiable nature of the energy function in Eq. (8), the closed-form solution for the optimal flows is not available. Therefore, we use an iterative method to solve the non-smooth optimization problem, which is friendly to back-propagation for training. We use the penalty method by introducing a quadratic penalty term for the constraint $\boldsymbol{Z}\mathbf{1}_n - \mathbf{1}_n = \mathbf{0}_n$ in Eq. (8), leading to

$$\min_{\boldsymbol{Z}} \underbrace{\frac{1}{2} \operatorname{Tr}\left[(\boldsymbol{R}^h \circ \boldsymbol{Z})\boldsymbol{Z}^T\right] + \frac{\alpha}{2}||\boldsymbol{Z}\mathbf{1}_n - \mathbf{1}_n||_2^2}_{H(\boldsymbol{Z})} + \underbrace{\lambda ||\boldsymbol{F}^h \circ \boldsymbol{Z}||_{1,1}}_{G(\boldsymbol{Z})}, \tag{9}$$

where $\alpha$ is the penalty strength parameter, and $G(\boldsymbol{Z})$ and $H(\boldsymbol{Z})$ denote the non-smooth part and the differentiable part, respectively. We then apply the proximal method Parikh & Boyd (2014) to iteratively solve the penalized problem, as outlined in appendix (A.2). To accelerate the convergence of the proximal iterations, we utilize the Barzilai-Borwein (BB) method for updating the step size (Barzilai & Borwein, 1988). Define $\boldsymbol{A}^{(k-1)} = \boldsymbol{Z}^{(k)} - \boldsymbol{Z}^{(k-1)}$ and $\boldsymbol{B}^{(k-1)} = \nabla_{\boldsymbol{Z}}^{(k)} H - \nabla_{\boldsymbol{Z}}^{(k-1)} H$. The proximal iteration can be expressed as:

$$\begin{cases} \boldsymbol{Y}^{(k)} = \boldsymbol{Z}^{(k)} - t^{(k)}\big(\boldsymbol{R}^h \circ \boldsymbol{Z}^{(k)} + \alpha(\boldsymbol{Z}^{(k)}\mathbf{1}_n - \mathbf{1}_n)\mathbf{1}_n^T\big), \\ \boldsymbol{Z}^{(k+1)} = \operatorname{sign}(\boldsymbol{Y}^{(k)}) \circ \max\big(|\boldsymbol{Y}^{(k)} - t^{(k)}\lambda\boldsymbol{F}^h|, \mathbf{0}_{n \times n}\big), \\ t^{(k)} = \frac{\langle \boldsymbol{A}^{(k-1)}, \boldsymbol{B}^{(k-1)} \rangle}{||\boldsymbol{B}^{(k-1)}||_F}, \end{cases} \tag{10}$$

where $|| \cdot ||_F$ denotes the Frobenius norm of a matrix and $\langle \cdot, \cdot \rangle$ denotes the Frobenius inner product of matrices. Since the elements of $\boldsymbol{R}^h$ and $\boldsymbol{F}^h$ are within the interval $(0, 1)$, the convergence of the proximal iteration method outlined in Eq. (10) can be ensured when $t^{(k)}$ is not greater than $(||\boldsymbol{R}^h||_2 + \alpha\sqrt{n})^{-1}$. See A.1 for the detailed derivation. The resulting optimal flows give rise to the SFi-attention pattern $\operatorname{SFi-ATT}^h(\boldsymbol{X}) = \boldsymbol{Z}^*(\boldsymbol{R}^h(\boldsymbol{X}), \boldsymbol{F}^h(\boldsymbol{X}))$.

## 4 THE SFI-FORMER ARCHITECTURE

In this section, we integrate the attention mechanisms from Sec. 3 into GTs and combine them with message-passing features of GNNs to enhance the ability to capture global information in graph data.

### 4.1 MOTIVATION FOR THE ARCHITECTURE

A graph representation learning task usually comes with a graph $\tilde{\mathcal{G}} = \{\tilde{\mathcal{V}}, \tilde{\mathcal{E}}\}$, with the node set $\tilde{\mathcal{V}} = \{\boldsymbol{v}_1, \boldsymbol{v}_2, \cdots, \boldsymbol{v}_n\}$ and the edges set $\tilde{\mathcal{E}} = \{\boldsymbol{e}_1, \boldsymbol{e}_2, \cdots, \boldsymbol{e}_m\}$. Note that $\tilde{\mathcal{G}}$ differs from the complete graph $\mathcal{G}$ for attention patterns introduced in Sec. 3.1. We denote the adjacency matrix of the graph $\tilde{\mathcal{G}}$ as $\boldsymbol{A} \in \mathbb{R}^{n \times n}$, and let $\boldsymbol{D}$ be the diagonal degree matrix where $\boldsymbol{D}_{i,i}$ represents the degree of node $\boldsymbol{v}_i$. Denote the $d$-dimensional feature vectors for all nodes as $\boldsymbol{X} \in \mathbb{R}^{n \times d}$.

Defining $\tilde{\boldsymbol{A}} = \hat{\boldsymbol{D}}^{-\frac{1}{2}}(\boldsymbol{A} + \boldsymbol{I})\hat{\boldsymbol{D}}^{-\frac{1}{2}}$, the message-passing step in Graph Convolutional Networks (GCN, Kipf & Welling (2016)) can be expressed as $\boldsymbol{X}^{(k+1)} = \sigma(\tilde{\boldsymbol{A}}\boldsymbol{X}^{(k)}\boldsymbol{W})$. In GCNs and many other message-passing-based GNNs, nodes are restricted to focus on 1-hop neighbors when constructing representations at each layer. As a result, multiple layers are needed to capture long-range interactions, but this approach is hindered by over-smoothing and over-squashing effects. In contrast, GTs can easily capture long-range dependencies of one node by its direct attentions to all others. However, not all information is relevant for downstream tasks; irrelevant or spurious patterns can propagate, even when having low attention scores. We believe that a more selective sparse attention can enable more effective and robust feature aggregation. Accordingly, we propose an architecture that combines SFi-attention with message-passing to achieve the best of both worlds.

### 4.2 Multi-head SFi-Attention Enhanced by Adjacency Components

Frameworks like GraphGPS already attempt to combine GTs with message-passing GNNs, but it remains unclear whether the GT module significantly contributes to model fitting. If it does, the resulting model may also inherit GTs' drawbacks, such as overemphasizing distant nodes (Xing et al., 2024). To alleviate the limitation within the GT module, we propose to enhance the GT with hard-wired adjacent connections as follows.

**Adjacency enhanced Attention:** Inspired by Resnet (He et al., 2016), we propose a residual-like learning approach, where the model learns the attention pattern beyond the features contributed from the adjacent nodes. The corresponding forward step is given by

$$\boldsymbol{X}^{(k+1)} = \boldsymbol{X}^{(k)} + (1+\gamma)^{-1}\sum_{h=1}^{H}[\tilde{\boldsymbol{A}} + \gamma\,\text{SFi-ATT}^h(\boldsymbol{X}^{(k)})]\boldsymbol{X}^{(k)}\boldsymbol{W}_V^h\boldsymbol{W}_O^h, \qquad (11)$$

where $\gamma$ is a learnable parameter that balances the contributions from the adjacency components and the SFi-attention. We will consider the following cases, each varying in hyper-parameter choices and methods for computing attention patterns.

- **Sparse Pattern:** SFi-attention is obtained by computing the optimal flows, i.e., $\text{SFi-ATT}^h(\boldsymbol{X}) = \boldsymbol{Z}^*(\boldsymbol{R}^h(\boldsymbol{X}), \boldsymbol{F}^h(\boldsymbol{X}))$, by following the procedures outlined in Sec. 3.3. Here, $\boldsymbol{R}^h(\boldsymbol{X})$ and $\boldsymbol{F}^h(\boldsymbol{X})$ are parameterized by two separate multi-head attention mechanisms as described in Sec. 3.3 as well. This process typically produces sparse attention patterns. We refer to the corresponding model in Eq. (11) as **SFi-Former**, which is illustrated in Figure 1.

- **Dense Pattern:** Similar to the above case, but with the $\ell_1$ penalty parameter $\lambda$ set to zero. In this case, $\text{SFi-ATT}(\cdot)$ effectively reduces to standard self-attention, resulting in dense attention patterns. For convenience, we refer to this special case as **DFi-Former**, which is also adjacency-enhanced. Recall that DFi-Former admits a closed-form solution, so iterative algorithms are not needed when computing the attention patterns.

DFi-Former is considered here for comparison with SFi-Former. It also serves as a pretrained model for initializing the parameters when computing $\boldsymbol{R}^h$ to accelerate training, where the resulting model is referred to as **SFi-Former+**.

## 5 Experiments

In this section, we evaluate our models across a wide range of graph datasets, including graph prediction, node prediction, and edge-level tasks. The results demonstrate that our models achieve state-of-the-art performance on many datasets, particularly those with long-range dependencies. We also conduct ablation studies to analyze how sparsity contributes to model performance and generalization.

**Our Models:** In this section, we propose three models for comparative experiment, all combining adjacency-enhanced SF-attention in Eq. (11) with message-passing GNNs in the GraphGPS framework (Rampasek et al., 2022). These models are: (i) SFi-Former, (ii) DFi-Former, and (iii) SFi-Former+, which initializes parameters using DFi-Former's checkpoint and computes attention patterns via iterative proximal methods as outlined in Sec. 3.3.

Table 1: Test performance on the LRGB dataset Dwivedi et al. (2022b). All models, except for ours, are categorized into three categories. The top group consists of GNN models based on local message passing, the middle group contains GTs, and the bottom group comprises sparse GT models and others. Results are presented as mean ± s.d. of 4 runs. The **first**, **second**, and **third** best are highlighted. **\***: Since the dataset COCO-SP is quite large, we only conduct a single run due to the limitation of computing resources.

| Model | COCO-SP | PascalVOC-SP | Peptides-Func | Peptides-Struct | PCQM-Contact |
|---|---|---|---|---|---|
| | F1 score ↑ | F1 score ↑ | AP ↑ | MAE ↓ | MRR ↑ |
| GCN | 0.0841 ± 0.0010 | 0.1268 ± 0.0060 | 0.5930 ± 0.0023 | 0.3496 ± 0.0013 | 0.3234 ± 0.0006 |
| GIN | 0.1339 ± 0.0044 | 0.1265 ± 0.0076 | 0.5498 ± 0.0079 | 0.3547 ± 0.0045 | 0.3180 ± 0.0027 |
| GatedGCN | 0.2641 ± 0.0045 | 0.2873 ± 0.0219 | 0.5864 ± 0.0077 | 0.3420 ± 0.0013 | 0.3242 ± 0.0011 |
| GAT | 0.1296 ± 0.0028 | 0.1753 ± 0.0329 | 0.5308 ± 0.0019 | 0.2731 ± 0.0402 | - |
| SPN | - | 0.2056 ± 0.0338 | 0.6926 ± 0.0247 | 0.2554 ± 0.0035 | - |
| SAN | 0.2592 ± 0.0158 | 0.3230 ± 0.0234 | 0.6439 ± 0.0064 | 0.2683 ± 0.0057 | 0.3350 ± 0.0003 |
| NAGphormer | 0.3458 ± 0.0070 | 0.4006 ± 0.0061 | - | - | - |
| GPS+Transformer | 0.3774 ± 0.0150 | 0.3689 ± 0.0131 | 0.6575 ± 0.0049 | 0.2510 ± 0.0015 | 0.3337 ± 0.0006 |
| NodeFormer | 0.3275 ± 0.0241 | 0.4015 ± 0.0082 | - | - | - |
| DIFFormer | 0.3620 ± 0.0012 | 0.3988 ± 0.0045 | - | - | - |
| GPS+BigBird | 0.2622 ± 0.0008 | 0.2762 ± 0.0069 | 0.5854 ± 0.0079 | 0.2842 ± 0.0139 | - |
| Exphormer | 0.3430 ± 0.0108 | 0.3975 ± 0.0043 | 0.6527 ± 0.0043 | 0.2481 ± 0.0007 | 0.3637 ± 0.0020 |
| Graph-mamba | 0.3909 ± 0.0128 | 0.4192 ± 0.0120 | 0.6972 ± 0.0100 | 0.2477 ± 0.0019 | - |
| DFi-Former | 0.3974 ± 0.0105 | 0.4400 ± 0.0113 | 0.6951 ± 0.0072 | 0.2470 ± 0.0034 | 0.3765 ± 0.0036 |
| SFi-Former | 0.3801* | 0.4737 ± 0.0096 | 0.6962 ± 0.0054 | 0.2478 ± 0.0029 | 0.3516 ± 0.0023 |
| SFi-Former+ | 0.3991* | 0.4670 ± 0.0071 | 0.7024 ± 0.0039 | 0.2467 ± 0.0026 | 0.3686 ± 0.0031 |

**Datasets:** We evaluated our models on the Long Range Graph Benchmark (Dwivedi et al., 2022b), including two image-based datasets (PascalVOC-SP, COCO-SP) and three molecular datasets (Peptides-Func, Peptides-Struct, and PCQM-Contact). We also performed evaluation on the Graph Neural Network Benchmark (Dwivedi et al., 2023), which includes two image-based datasets (CIFAR10, MNIST) and two synthetic SBM datasets (PATTERN, CLUSTER).

**Baselines:** We evaluate the performance of SFi-Former by comparing it with basic message-passing GNNs (MPNNs), GTs, and other competitive graph neural networks. For basic MPNNs, we consider models such as GCN (Kipf & Welling, 2016), GIN (Xu et al., 2018), GAT Veličković et al. (2017), SPN (Abboud et al., 2022), GraphSAGE (Hamilton et al., 2017), along with their enhanced versions (e.g. Gated-GCN (Bresson & Laurent, 2017)). For GTs, we include recent competitive models such as SAN (Kreuzer et al., 2021a), NAGphormer (Chen et al., 2022b), GPS-Transformer (Rampasek et al., 2022), as well as sparse GTs like Performer, BigBird (Zaheer et al., 2020) and Exphormer (Shirzad et al., 2023). Furthermore, we compare against other competitive graph neural networks, including Graph-mamba (Behrouz & Hashemi, 2024) and DIFFormer (Wu et al., 2023).

**Setup:** We conducted our experiments within the GraphGPS framework proposed by (Rampasek et al., 2022). All experiments were run on Nvidia A100 GPUs with 40GB memory and Nvidia A6000 GPUs with 48GB memory. Model parameters are provided in Appendix A.4.2.

## 5.1 LONG RANGE GRAPH BENCHMARK

Table 1 presents the results of our models on the Long-Range Graph Benchmark (LRGB) Dwivedi et al. (2022b), which consists of five challenging datasets designed to assess a model's ability to capture long-range interactions (LRI) in graphs. Our models demonstrate superior performance, surpassing all existing models on these long-range datasets. Notably, they significantly outperform previous best results on the PascalVOC-SP and COCO-SP datasets.

Remind that in DFi-Former, the adjacency-enhanced method is applied to the standard-attention mechanism. It already demonstrates competitive performance, which indicates the effectiveness of our design of the adjacency-enhanced method. Further improvements in test performance are observed with SFi-Former and SFi-Former+ across most datasets (except PCQM-Contact), highlighting the effectiveness of the proposed SFi-attention mechanism and its associated iterative computation methods.

Table 2: Test performance on the GNNbenchmark dataset Dwivedi et al. (2023). Results are presented as mean ± s.d. of 4 runs. The **first**, **second**, and **third** best are highlighted.

| Model | MNIST | CIFAR-10 | PATTERN | CLUSTER |
|---|---|---|---|---|
| | Accuracy ↑ | Accuracy ↑ | Accuracy ↑ | Accuracy ↑ |
| GCN | 0.9071 ± 0.0021 | 0.5571 ± 0.0038 | 0.7189 ± 0.0033 | 0.6850 ± 0.0098 |
| GIN | 0.9649 ± 0.0025 | 0.5526 ± 0.0153 | 0.8539 ± 0.0013 | 0.6472 ± 0.0155 |
| GatedGCN | 0.9734 ± 0.0014 | 0.6731 ± 0.0031 | 0.8557 ± 0.0008 | 0.7384 ± 0.0033 |
| GAT | 0.9554 ± 0.0021 | 0.6422 ± 0.0046 | 0.7827 ± 0.0019 | 0.7059 ± 0.0045 |
| GraphSAGE | 0.9731 ± 0.0009 | 0.6577 ± 0.0030 | 0.5049 ± 0.0001 | - |
| SAN | - | - | 0.8658 ± 0.0004 | 0.7669 ± 0.0065 |
| GPS+Transformer | 0.9811 ± 0.0011 | 0.7226 ± 0.0031 | 0.8664 ± 0.0011 | 0.7802 ± 0.0018 |
| GPS+BigBird | 0.9817 ± 0.0001 | 0.7048 ± 0.0011 | 0.8600 ± 0.0014 | - |
| Exphormer | **0.9855 ± 0.0003** | **0.7469 ± 0.0013** | 0.8670 ± 0.0003 | 0.7807 ± 0.0002 |
| Graph-mamba | 0.9839 ± 0.0018 | **0.7456 ± 0.0038** | **0.8709 ± 0.0126** | - |
| DFi-Former | **0.9848 ± 0.0005** | 0.7391 ± 0.0045 | 0.8641 ± 0.0011 | **0.7820 ± 0.0012** |
| SFi-Former | **0.9846 ± 0.0009** | 0.7366 ± 0.0058 | **0.8674 ± 0.0017** | **0.7828 ± 0.0015** |
| SFi-Former+ | 0.9831 ± 0.0012 | **0.7459 ± 0.0053** | **0.8678 ± 0.0021** | **0.7810 ± 0.0011** |

In particular, our models show significant performance improvements on the COCO-SP and PascalVOC-SP (C&P) datasets, while the gains on the other three datasets are comparatively modest. A possible explanation for this disparity is that the nodes in the C&P datasets represent superpixels from images, where many background nodes don't require interactions and not all of them contribute to the semantically relevant nodes. In these cases, a sparse attention pattern effectively captures relevant interactions, boosting performance. In contrast, the Peptides and PCQM datasets consist of atom-based nodes, where all nodes may hold similar importance, diminishing the benefit of sparsity. This is further supported by our investigation: in the C&P datasets, around 20% of node interactions receive zero attention, compared to only 5% in the other datasets.

## 5.2 GNN Benchmark Datasets

Table 2 showcases the performance of our models in GNN benchmark datasets (Dwivedi et al., 2023). The results demonstrate that our models not only excel at handling long-range dependency challenges but also perform effectively in general graph learning tasks.

## 5.3 Ablation Studies

In this section, we conduct a series of ablation studies. First, to assess the contribution of each component in SFi-Former, we separately tested the impact of the adjacency-enhanced method and the sparse attentions on the results. Neither component alone yielded the most competitive results, highlighting the importance of both in enhancing prediction performance. Second, we explored the optimal parameters for the flow network's energy framework. The results show that no single parameter set consistently outperformed others across all datasets, but our model exhibited strong potential under well-tuned conditions. Based on this ablation studies, we select $\lambda^* = 1.0$ and $\alpha = 0.1$ as the parameters for our models, as they provided consistently optimal performance across datasets.

## 5.4 Role of Sparsity in Enhancing Generalization

Beyond the prediction performance on test datasets, it is also crucial to evaluate the gap between training and testing metrics, as this provides insights into the model's generalization ability. A larger train-test gap typically suggests a higher risk of over-fitting. As outlined in previous sections, our adaptive sparse attention mechanism is expected to be more selective in feature aggregation, leading to models that are more stable and generalizable. To demonstrate this, we plot the train-test gap of SFi-Former across three datasets and compare it with the GraphGPS model using dense attention. The results are shown in Figure 3. Specifically, in the PascalVOC-SP and Peptides-Func datasets, F1-score and accuracy have been used as evaluation metrics (the larger the better). Consequently, a smaller gap between the training and testing metrics implies less over-fitting to the training data. Figures 3a and 3b indicate that SFi-Former has a train-test smaller gap compared to GraphGPS. On

Table 3: Ablation Studies. We analyze the impact of each component in our models as follows: (i) The penalty coefficient $\alpha$ for the flow conservation constraint, as introduced in Eq. (9). (ii) The adjacency-enhanced attention mechanism as described in Eq. (11). In this table, $\tilde{A}$ refers to the application of the adjacency-enhanced method, while SP denotes the use of SFi-attention. (iii) The hyperparameter $\lambda^*$ which balances friction and resistance terms in Eq. (8). In the table, $\lambda^*/N^*$ corresponds to $\lambda$ in Eq. (8), where $N^*$ is the maximum number of nodes in a batch.

| Model | $\lambda^*$ | $\alpha$ | Attention | PascalVOC-SP | Peptides-Func | Peptides-Struct |
|---|---|---|---|---|---|---|
| | | | | F1 score ↑ | AP ↑ | MAE ↓ |
| GPS | - | - | - | $0.3748 \pm 0.0109$ | $0.6535 \pm 0.0023$ | $\textbf{0.2510} \pm \textbf{0.0023}$ |
| SFi-Former-$\lambda_1$ | 0.5 | 0.1 | SP + $\tilde{A}$ | $\textbf{0.4853} \pm \textbf{0.0057}$ | $\textbf{0.6983} \pm \textbf{0.0034}$ | $0.2527 \pm 0.0013$ |
| SFi-Former-$\lambda_2$ | 2.0 | 0.1 | SP + $\tilde{A}$ | $0.4583 \pm 0.0066$ | $0.6844 \pm 0.0065$ | $0.2517 \pm 0.0016$ |
| SFi-Former-$\lambda_3$ | 5.0 | 0.1 | SP + $\tilde{A}$ | $\textbf{0.4839} \pm \textbf{0.0073}$ | $\textbf{0.7025} \pm \textbf{0.0019}$ | $0.2528 \pm 0.0011$ |
| SFi-Former-$\alpha_1$ | 1.0 | 0.01 | SP + $\tilde{A}$ | $0.4600 \pm 0.0083$ | $0.6851 \pm 0.0015$ | $0.2529 \pm 0.0018$ |
| SFi-Former-$\alpha_2$ | 1.0 | 0.5 | SP + $\tilde{A}$ | $0.4581 \pm 0.0052$ | $\textbf{0.7006} \pm \textbf{0.0027}$ | $\textbf{0.2504} \pm \textbf{0.0034}$ |
| SFi-Former-$\alpha_3$ | 1.0 | 1.0 | SP + $\tilde{A}$ | $0.4713 \pm 0.0076$ | $0.6873 \pm 0.0042$ | $0.2552 \pm 0.0026$ |
| SFi-Former | 1.0 | 0.1 | SP + $\tilde{A}$ | $\textbf{0.4737} \pm \textbf{0.0096}$ | $0.6962 \pm 0.0054$ | $\textbf{0.2478} \pm \textbf{0.0029}$ |
| SFi-Former-SP | 1.0 | 0.1 | SP | $0.4522 \pm 0.0079$ | $0.6766 \pm 0.0054$ | $0.2520 \pm 0.0017$ |
| SFi-Former-$\tilde{A}$ | 1.0 | 0.1 | $\tilde{A}$ | $0.3800 \pm 0.0091$ | $0.6552 \pm 0.0065$ | $0.2511 \pm 0.0035$ |

the other hand, the Peptides-Struct dataset utilizes MAE as an evaluation metric (the smaller the better), so we plot the negative train-test gap, and the result in Figure 3c demonstrates that SFi-Former is also better than GraphGPS. In summary, SFi-Former consistently exhibits a smaller train-test gap than the GraphGPS using dense attention, which indicates that SFi-Former is less prone to over-fitting and highlights its superior generalization ability.

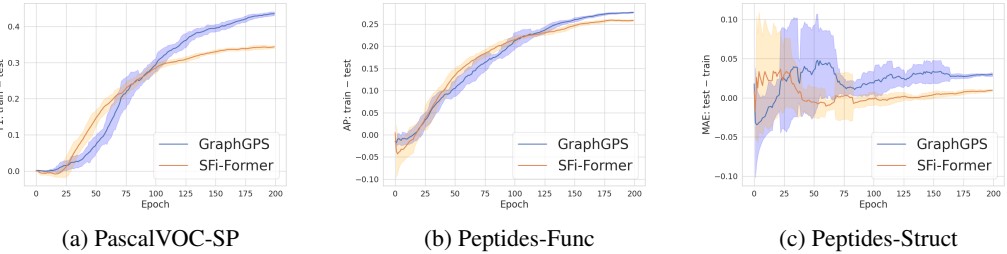

(a) PascalVOC-SP  (b) Peptides-Func  (c) Peptides-Struct

Figure 3: Differences between the training and testing metrics for the GraphGPS and SFi-Former models throughout the entire training process across three datasets. Models with smaller differences between these metrics indicate better generalization.

## 6 CONCLUSION

In this paper, we introduce SFi-Former, a novel graph transformer architecture featuring a sparse attention mechanism that selectively aggregates features from other nodes through adaptable sparse attention. The sparse attention patterns in SFi-Former correspond to optimal network flows derived from an energy-minimization problem, offering an interesting electric-circuit interpretation of the standard self-attention mechanism (as a special case of our framework). This framework also provides flexibility for extending to other innovative attention mechanisms by adjusting the energy function and related components. Further augmented by an adjacency-enhanced method, SFi-Former is able to balance local message-passing and global attetion within the graph transformer module, effectively capturing long-range interactions across various graph datasets and achieving state-of-the-art performance. Additionally, SFi-Former shows smaller train-test gaps, demonstrating reduced susceptibility to overfitting. We envisage that SFi-Former and the proposed flow-based energy minimization framework hold promise for future research in other areas of machine learning.

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

# A  APPENDIX

## A.1  PROOFS

### A.1.1  PRELIMINARIES

This section introduces the convergence of approximate point gradient descent methods. When proposing an approximate gradient descent algorithm, we require $f(X)$ to be a convex function during the convergence analysis.

**Definition 1**(Proximal Operator).

To provide clarity, we first define the proximity operator for a convex function. Let the function $h : \mathbb{R}^{n \times n} \to \mathbb{R}^{n \times n}$ be defined as follows:

$$\text{prox}_h(\boldsymbol{X}) = \arg\min_{\boldsymbol{U}} \left( h(\boldsymbol{U}) + \frac{1}{2}\|\boldsymbol{U} - \boldsymbol{X}\|^2 \right) \quad \boldsymbol{U} \in \mathbb{R}^{n \times n}$$

Using the Weierstrass theorem, we can guarantee that $h(\boldsymbol{U})$ has a minimizer within a bounded domain. Since the minimizer exists, the proximity operator is well-defined. If $h$ is a proper closed convex function, then for any $X$, the value of $\text{prox}(X)$ exists and is unique.

**Theorem 1**(Relationship between Proximal Operators and Subgradients). If $\boldsymbol{U}$ is the optimal point,

$$\boldsymbol{U} = \text{prox}_h(\boldsymbol{X}) \iff \boldsymbol{X} - \boldsymbol{U} \in \partial h(\boldsymbol{U})$$

where $\partial h(\boldsymbol{U})$ is the subgradient of function $h$. **Proof**: If $\boldsymbol{U} = \text{prox}_h(\boldsymbol{X})$, the optimality condition is given by:

$$0 \in \partial h(\boldsymbol{U}) + (\boldsymbol{U} - \boldsymbol{X}), \quad \text{so} \quad \boldsymbol{X} - \boldsymbol{U} \in \partial h(\boldsymbol{U})$$

Conversely, if $\boldsymbol{X} - \boldsymbol{U} \in \partial h(\boldsymbol{U})$, by the definition of the subgradient,

$$h(\boldsymbol{V}) \geq h(\boldsymbol{U}) + \langle \boldsymbol{X} - \boldsymbol{U}, \boldsymbol{V} - \boldsymbol{U} \rangle, \quad \forall \boldsymbol{V} \in \mathbb{R}^{n \times n}$$

Adding $\frac{1}{2}\|\boldsymbol{V} - \boldsymbol{X}\|^2$ to both sides,

$$h(\boldsymbol{V}) + \frac{1}{2}\|\boldsymbol{V} - \boldsymbol{X}\|^2 \geq h(\boldsymbol{U}) + \frac{1}{2}\|\boldsymbol{U} - \boldsymbol{X}\|^2, \quad \forall \boldsymbol{V} \in \mathbb{R}^{n \times n}$$

Thus, we have $\boldsymbol{U} = \text{prox}_h(\boldsymbol{X})$.

Using $th$ as a substitution for $h$, the conclusion can be rewritten as:

$$\boldsymbol{U} = \text{prox}_{th}(\boldsymbol{X}) \iff \boldsymbol{U} = \boldsymbol{X} - t \cdot \partial h(\boldsymbol{U})$$

**Assumption 1** (Lipschitz condition).

1. $f : \mathbb{R}^{n \times n} \to \mathbb{R}^1$ is differentiable.
2. Let $\text{prox}_h$ denote the proximal operator of convex function $h : \mathbb{R}^{n \times n} \to \mathbb{R}^1$. The definition of $\text{prox}_h$ is reasonable.
3. $\psi(\boldsymbol{X}) = f(\boldsymbol{X}) + h(\boldsymbol{X})$ has a bounded minimum $\psi^*$, and at point $\boldsymbol{X}^*$, it attains its minimum.

Moreover, the gradient $\nabla_{\boldsymbol{X}} f(X)$ satisfies the Lipschitz condition. i.e., for some constant $L$, we have

$$\|\nabla_{\boldsymbol{X}} f(\boldsymbol{X}) - \nabla_{\boldsymbol{Y}} f(\boldsymbol{Y})\| \leq L\|\boldsymbol{X} - \boldsymbol{Y}\|, \quad \forall \boldsymbol{X}, \boldsymbol{Y} \in \mathbb{R}^{n \times n}.$$

According to **Assumption 1**, $\psi(\boldsymbol{X})$ consists of two components: for the convex part $f$, we solve the problem using gradient descent, and for the part $h$, we utilize the proximal operator. Thus, the iteration formula can be derived as follows:

$$\boldsymbol{X}^{k+1} = \text{prox}_{t^k, h}(\boldsymbol{X}^k - t_k \nabla_{\boldsymbol{X}} f(\boldsymbol{X}^k)) \tag{Iteration}$$

The above conditions ensure the convergence results of the approximate gradient method: In the case of a fixed step size $t_k = t \in \left(0, \frac{1}{L}\right]$, the function value at point $x^k$, $\psi(x^k)$, converges to $\psi^*$ at a rate of $O\left(\frac{1}{k}\right)$. Before formally presenting the convergence result, we first introduce a new function.

**Definition 2**(Gradient Mapping). Let $f(\boldsymbol{X})$ and $h(\boldsymbol{X})$ satisfy **Assumption 1**, and let $t > 0$ be a constant. We define the gradient mapping $G_t : \mathbb{R}^{n \times n} \to \mathbb{R}^{n \times n}$ as follows:

$$G_t(\boldsymbol{X}) = \frac{1}{t} \left( \boldsymbol{X} - \mathrm{prox}_{th}(\boldsymbol{X} - t\nabla_{\boldsymbol{X}} f(\boldsymbol{X})) \right).$$

It can be shown that $G_t(\boldsymbol{X})$ functions as the 'search direction' for each iteration in the approximate gradient method, i.e.

$$\boldsymbol{X}^{k+1} = \mathrm{prox}_{th}(\boldsymbol{X}^k - t\nabla_{\boldsymbol{X}} f(\boldsymbol{X}^k)) = \boldsymbol{X}^k - tG_t(\boldsymbol{X}^k).$$

Notably, $G_t(\boldsymbol{X})$ is not the gradient or subgradient of $\psi = f + h$. The relationship between the gradient and the subgradient can be derived as follows:

$$G_t(\boldsymbol{X}) - \nabla_{\boldsymbol{X}} f(\boldsymbol{X}) \in \partial h(\boldsymbol{X} - tG_t(\boldsymbol{X})).$$

Additionally, as $G_t(\boldsymbol{X})$ serves as the "search direction," its relationship with the convergence of the algorithm is critical. In fact, $G_t(\boldsymbol{X}) = 0$ at the minimum of $\psi(\boldsymbol{X}) = f(\boldsymbol{X}) + h(\boldsymbol{X})$.

Based on the above definition, we now introduce the convergence of the approximate gradient method.

**Theorem 2**: Under **Assumption 1**, with a fixed step size $t_k = t \in \left(0, \frac{1}{L}\right]$, the sequence generated by equation (Iteration) satisfies:

$$\psi(\boldsymbol{X}^k) - \psi^* \leq \frac{1}{2kt} \|\boldsymbol{X}^0 - \boldsymbol{X}^*\|^2.$$

**Proof**: By applying the Lipschitz continuity property from **Assumption 1** along with the quadratic upper bound, we have:

$$f(\boldsymbol{Y}) \leq f(\boldsymbol{X}) + \nabla_{\boldsymbol{X}} f(\boldsymbol{X})^T(\boldsymbol{Y} - \boldsymbol{X}) + \frac{L}{2} \|\boldsymbol{Y} - \boldsymbol{X}\|^2, \quad \forall \boldsymbol{X}, \boldsymbol{Y} \in \mathbb{R}^{n \times n}.$$

Let $\boldsymbol{Y} = \boldsymbol{X} - tG_t(\boldsymbol{X})$,

$$f(\boldsymbol{X} - tG_t(\boldsymbol{X})) \leq f(\boldsymbol{X}) - t\nabla_{\boldsymbol{X}} f(\boldsymbol{X})^T G_t(\boldsymbol{X}) + \frac{t^2 L}{2} \|G_t(\boldsymbol{X})\|^2.$$

For $0 < t \leq \frac{1}{L}$,

$$f(\boldsymbol{X} - tG_t(\boldsymbol{X})) \leq f(\boldsymbol{X}) - t\nabla_{\boldsymbol{X}} f(\boldsymbol{X})^T G_t(\boldsymbol{X}) + \frac{t}{2} \|G_t(\boldsymbol{X})\|^2.$$

Moreover, since $f(\boldsymbol{X}), h(\boldsymbol{X})$ are convex functions, for any $\boldsymbol{Z} \in \mathbb{R}^{n \times n}$,

$$h(\boldsymbol{Z}) \geq h(\boldsymbol{X} - tG_i(\boldsymbol{X})) + (G_i(\boldsymbol{X}) - \nabla f(\boldsymbol{X}))^T(\boldsymbol{Z} - \boldsymbol{X} + tG_i(\boldsymbol{X})),$$

$$f(\boldsymbol{Z}) \leq f(\boldsymbol{X}) + \nabla f(\boldsymbol{X})^T(\boldsymbol{Z} - \boldsymbol{X}).$$

The inequality regarding $h(\boldsymbol{Z})$ uses the relationship (8.1.10). By simplifying, we obtain:

$$h(\boldsymbol{X} - tG_i(\boldsymbol{X})) \leq h(\boldsymbol{Z}) - (G_i(\boldsymbol{X}) - \nabla_{\boldsymbol{X}} f(\boldsymbol{X}))^T(\boldsymbol{Z} - \boldsymbol{X}) + \frac{1}{2} \|G_i(\boldsymbol{X})\|^2.$$

We get for any $\boldsymbol{Z} \in \mathbb{R}^{n \times n}$ in the global inequality that:

$$\psi(\boldsymbol{X} - tG_i(\boldsymbol{X})) \leq \psi(\boldsymbol{Z}) + G_i(\boldsymbol{Z})^T(\boldsymbol{X} - \boldsymbol{Z}) - \frac{t}{2} \|G_i(\boldsymbol{X})\|^2.$$

Therefore, for each step of the iteration,

$$\boldsymbol{X} = \boldsymbol{X} - tG_i(\boldsymbol{X}),$$

In the global inequality, taking $z = x^*$,

$$\psi(\boldsymbol{X}^t) - \psi(\boldsymbol{X}^*) \leq G_i(\boldsymbol{X})^T(\boldsymbol{X}^t - \boldsymbol{X}^*) - \frac{t}{2} \|G_i(\boldsymbol{X})\|^2.$$

This simplifies to:

$$= \frac{1}{2t} \left( \|\boldsymbol{X} - \boldsymbol{X}^*\|^2 - \|\boldsymbol{X}^t - \boldsymbol{X}^*\|^2 - \|\boldsymbol{X} - tG_i(\boldsymbol{X}) - \boldsymbol{X}^*\|^2 \right)$$

which leads to:

$$= \frac{1}{2t} \left( \|\boldsymbol{X}^0 - \boldsymbol{X}^*\|^2 - \|\boldsymbol{X}^t - \boldsymbol{X}^*\|^2 \right).$$

Summing for $i = 1, 2, \ldots, k$,

$$\sum_{i=1}^{k} \left( \psi(\boldsymbol{X}^i) - \psi(\boldsymbol{X}^*) \right) \le \frac{1}{2t} \sum_{i=1}^{k} \left( \|\boldsymbol{X}^{i-1} - \boldsymbol{X}^*\|^2 - \|\boldsymbol{X}^i - \boldsymbol{X}^*\|^2 \right)$$

$$= \frac{1}{2t} \left( \|\boldsymbol{X}^0 - \boldsymbol{X}^*\|^2 - \|\boldsymbol{X}^k - \boldsymbol{X}^*\|^2 \right)$$

Thus,

$$\psi(\boldsymbol{X}^k) - \psi(\boldsymbol{X}^*) \le \frac{1}{2kt} \|\boldsymbol{X}^0 - \boldsymbol{X}^*\|^2.$$

According to **Theorem 2**, the requirement for convergence is that the step size must be no more than to the inverse of the Lipschitz constant $L$ corresponding to $\nabla_{\boldsymbol{X}} f$.

### A.1.2 Convergence Analysis

**Definition 3** Optimization energy function. The formal optimization energy function of $h^{th}$ head $E^h(\boldsymbol{Z}; \boldsymbol{R}^h, \boldsymbol{F}^h) : \mathbb{R}^{n \times n} \to \mathbb{R}$ is defined as follows:

$$E^h(\boldsymbol{Z}; \boldsymbol{R}^h, \boldsymbol{F}^h) = \frac{1}{2} \operatorname{Tr} \left[ (\boldsymbol{R}^h \circ \boldsymbol{Z}) \boldsymbol{Z}^T \right] + \lambda \|\boldsymbol{F}^h \circ \boldsymbol{Z}\|_{1,1} + \frac{\alpha}{2} \|\boldsymbol{Z} \boldsymbol{1}_n - \boldsymbol{1}_n\|^2,$$

$$\psi(\boldsymbol{Z}) = f(\boldsymbol{Z}) + h(\boldsymbol{Z}), h(\boldsymbol{Z}) = \lambda \|\boldsymbol{F}^h \circ \boldsymbol{Z}\|_{1,1}.$$

**Proposition** (Constraint of step size $t^h$ in proximal optimization Barzilai & Borwein (1988)). The function value of the algorithm at the iteration point $X^k$, denoted as $\phi(X^k)$, converges to $\phi(X^*)$ at a rate of $o(1/k)$, when the following condition is satisfied in $h^{th}$ head:

$$0 < t^h \le \frac{1}{\|\boldsymbol{R}^h\| + \alpha \sqrt{n}}.$$

Moreover,

$$0 < t^h \le \frac{1}{\alpha \sqrt{n} + 1}.$$

Each row component of matrix $\boldsymbol{R}^h$ satisfies:

$$\sum_{j=1}^{n} \boldsymbol{R}_{ij}^h = 1, \forall i = 1, 2, \cdots, n.$$

This is because the matrix $\boldsymbol{R}^h$ represents the attention between query and key nodes. According to Perron-Frobenius theorem, the non-expansive feature of the attention matrix introduces $\lambda_{max, \boldsymbol{R}^h} = 1$, which denotes $\|\boldsymbol{R}^h\| \le 1$. So we can guarantee the convergence of the optimal algorithm by taking $0 < t^h \le \frac{1}{\alpha \sqrt{n} + 1}$, which provides an efficient method to setup the iteration step $t^h$ for given $\lambda$ and $\alpha$ before the training starts.

**Proof.** According to **Theorem 2**, the algorithm is convergent if $0 < t < \frac{1}{L_f}$, where $L_f$ is the convex part $f(\boldsymbol{X})$ of the optimal function $\psi(\boldsymbol{X})$. Notice that the function $f(\boldsymbol{X})$ satisfies $\|\nabla_{\boldsymbol{X}} f(\boldsymbol{X}) - \nabla_{\boldsymbol{Y}} f(\boldsymbol{Y})\| \le L_f \|\boldsymbol{X} - \boldsymbol{Y}\|, \forall \boldsymbol{X}, \boldsymbol{Y} \in \mathbb{R}^{n \times n}$. We can derive :

$$L_f = \sup_{\boldsymbol{X}, \boldsymbol{Y}} \frac{\|\nabla_{\boldsymbol{X}} f(\boldsymbol{X}) - \nabla_{\boldsymbol{Y}} f(\boldsymbol{Y})\|}{\|\boldsymbol{X} - \boldsymbol{Y}\|}.$$

For each row component of $E^h(\boldsymbol{Z}; \boldsymbol{R}^h, \boldsymbol{F}^h)$ in **Definition 3**, we have

$$E_i^h(\boldsymbol{Z}_{i,:}; \boldsymbol{R}_{i,:}^h, \boldsymbol{F}_{i,:}^h) = \frac{1}{2} \operatorname{Tr} \left[ (\boldsymbol{R}_{i,:}^h \circ \boldsymbol{Z}_{i,:}) \boldsymbol{Z}_{i,:}^T \right] + \lambda \|\boldsymbol{F}_{i,:}^h \circ \boldsymbol{Z}_{i,:}\|_1 + \frac{\alpha}{2} \|\boldsymbol{Z}_{i,:} \boldsymbol{1}_n - 1\|^2,$$

$$E^h = \sum_{i=1}^{n} E_i^h, \forall i = 1, 2, \cdots, n.$$

The convex part of $E_i^h$ is defined as follow:

$$f_i^h(\boldsymbol{Z}_{i,:}; \boldsymbol{R}_{i,:}^h, \boldsymbol{F}_{i,:}^h) = \frac{1}{2}\operatorname{Tr}\left[(\boldsymbol{R}_{i,:}^h \circ \boldsymbol{Z}_{i,:})\boldsymbol{Z}_{i,:}^T\right] + \frac{\alpha}{2}\|\boldsymbol{Z}_{i,:}\boldsymbol{1}_n - 1\|^2.$$

Let $\boldsymbol{z}_i = \boldsymbol{Z}_{i,:}$ denotes the $i-th$ row component,

$$\nabla_{\boldsymbol{z}_i} f_i(\boldsymbol{z}_i) = \boldsymbol{R}_{i,:}^h \boldsymbol{z}_i + \alpha(\boldsymbol{z}_i\boldsymbol{1}_n - 1).$$

Then

$$\begin{aligned}
\|\nabla_{\boldsymbol{x}_i} f_i^h(\boldsymbol{x}_i) - \nabla_{\boldsymbol{y}_i} f_i^h(\boldsymbol{y}_i)\| &= \|\boldsymbol{R}_{i,:}^h(\boldsymbol{x}_i - \boldsymbol{y}_i) + \alpha(\boldsymbol{x}_i - \boldsymbol{y}_i)\boldsymbol{1}_n\| \\
&\leq \|\boldsymbol{R}_{i,:}^h(\boldsymbol{x}_i - \boldsymbol{y}_i)\| + \alpha\|(\boldsymbol{x}_i - \boldsymbol{y}_i)\boldsymbol{1}_n\| \\
&\leq (\|\boldsymbol{R}_{i,:}^h\| + \alpha\|\boldsymbol{1}_n\|)\|\boldsymbol{x}_i - \boldsymbol{y}_i\| \\
&= (\|\boldsymbol{R}_{i,:}^h\| + \alpha\sqrt{n})\|\boldsymbol{x}_i - \boldsymbol{y}_i\| \forall i = 1, 2, \cdots n
\end{aligned}$$

$$\begin{aligned}
\|\nabla_{\boldsymbol{X}} f^h(\boldsymbol{X}) - \nabla_{\boldsymbol{Y}} f^h(\boldsymbol{Y})\| &= \left\|\begin{matrix} \nabla_{\boldsymbol{x}_1} f_1^h(\boldsymbol{x}_1) - \nabla_{\boldsymbol{y}_1} f_1^h(\boldsymbol{y}_1) \\ \nabla_{\boldsymbol{x}_2} f_2^h(\boldsymbol{x}_2) - \nabla_{\boldsymbol{y}_2} f_2^h(\boldsymbol{y}_2) \\ \vdots \\ \nabla_{\boldsymbol{x}_n} f_n^h(\boldsymbol{x}_n) - \nabla_{\boldsymbol{y}_n} f_n^h(\boldsymbol{y}_n) \end{matrix}\right\| \\
&\leq \left\|\begin{matrix} (\|\boldsymbol{R}_{1,:}^h\| + \alpha\sqrt{n})(\boldsymbol{x}_1 - \boldsymbol{y}_1) \\ (\|\boldsymbol{R}_{2,:}^h\| + \alpha\sqrt{n})(\boldsymbol{x}_2 - \boldsymbol{y}_2) \\ \vdots \\ (\|\boldsymbol{R}_{n,:}^h\| + \alpha\sqrt{n})(\boldsymbol{x}_n - \boldsymbol{y}_n) \end{matrix}\right\| \\
&\leq (\{\|\boldsymbol{R}_{i,:}^h\|\}_{max} + \alpha\sqrt{n})\left\|\begin{matrix} \boldsymbol{x}_1 - \boldsymbol{y}_1 \\ \boldsymbol{x}_2 - \boldsymbol{y}_2 \\ \vdots \\ \boldsymbol{x}_n - \boldsymbol{y}_n \end{matrix}\right\| \\
&\leq (\|\boldsymbol{R}^h\| + \alpha\sqrt{n})\|\boldsymbol{X} - \boldsymbol{Y}\|
\end{aligned}$$

Thus $L_f = \|\boldsymbol{R}^h\| + \alpha\sqrt{n}$. According to **Theorem 2**, the algorithm is convergent when $0 < t \leq \frac{1}{L_f}$.

## A.2 PROXIMAL METHOD FOR NON-SMOOTH OPTIMIZATION

Consider the following optimization problem

$$\min_{\boldsymbol{Z}} E(\boldsymbol{Z}) = H(\boldsymbol{Z}) + G(\boldsymbol{Z}), \tag{12}$$

where $H(\cdot)$ is a smooth function, and $G(\cdot)$ is a non-smooth function. The proximal method for solving this problem involves iterating the following steps

$$\begin{cases} \boldsymbol{Y}^{(k)} = \boldsymbol{Z}^{(k)} - t^{(k)}\nabla_{\boldsymbol{Z}}^{(k)} H \\ \boldsymbol{Z}^{(k+1)} = \operatorname{prox}_{t^{(k)},G}(\boldsymbol{Y}^{(k)}) \\ t^{(k+1)} = u(t^{(k)}) \end{cases} \tag{13}$$

where $u(\cdot)$ is a function used to update the step size $t$.

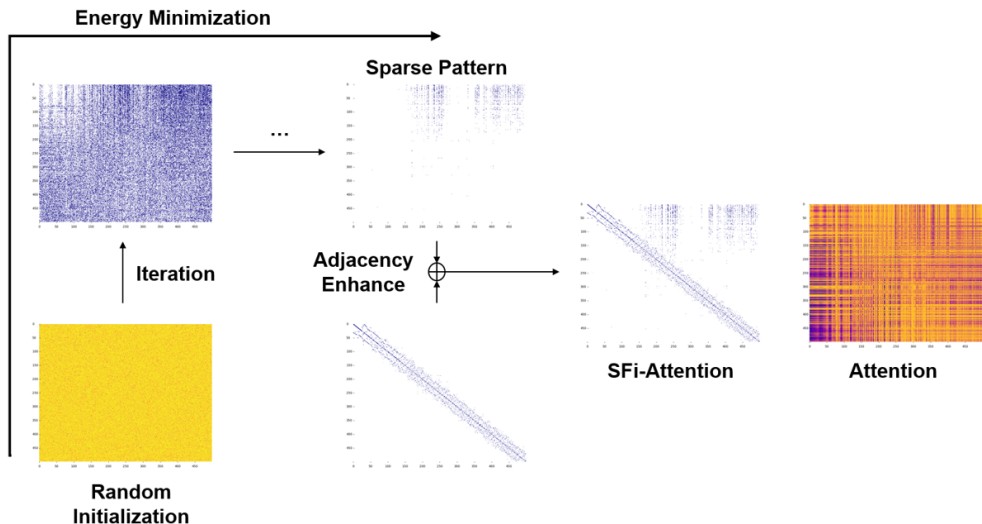

Figure 4: Demonstration of SFi-attention and its iterative process. We utilize a logarithmic transformation on the original attention values, represented with a viridis colorbar, where yellow areas indicate values near 1 and blue areas signify values close to 0 but above the threshold of 1e-8. Values exceptionally close to (below the threshold) appear white in this representation.

## A.3 DEMONSTRATION OF SPARSITY IN SFI-ATTENTION

To verify the true sparse ability of energy flow mechanism, we employ a series of transforms to visually present the intuitive distribution of attention in Figure 4. Concurrently, we also visualize how the attention adjusted during the energy function minimization process, the adjacency enhancement, and compare the final attention results of DFi-Former and SFi-Former. From the figure, we observe that the attention matrix obtained by the SFi-Former is indeed very sparse, with only a small portion of the features being captured. The final SFi-attention we obtain has values close to 0 compared to vanilla Attention. The influence of the matrix $\tilde{A}$ also becomes a crucial part after the adjacency enhancement.

## A.4 EXPERIMENTAL DETAILS

### A.4.1 DATASET DESCRIPTION

Table 4: Overview of the graph learning dataset (Dwivedi et al., 2023; 2022b) used in this study.

| Dataset | Graphs | Avg. nodes | Avg. edges | Directed | Prediction level | Prediction task | Metric |
|---|---|---|---|---|---|---|---|
| MNIST | 70,000 | 70.6 | 564.5 | Yes | graph | 10-class classif. | Accuracy |
| CIFAR10 | 60,000 | 117.6 | 941.1 | Yes | graph | 10-class classif. | Accuracy |
| PATTERN | 14,000 | 118.9 | 3,039.3 | No | inductive node | binary classif. | Accuracy |
| CLUSTER | 12,000 | 117.2 | 2,150.9 | No | inductive node | 6-class classif. | Accuracy |
| PascalVOC-SP | 11,355 | 479.4 | 2,710.5 | No | inductive node | 21-class classif. | F1 score |
| COCO-SP | 123,286 | 476.9 | 2,693.7 | No | inductive node | 81-class classif. | F1 score |
| PCQM-Contact | 529,434 | 30.1 | 61.0 | No | inductive link | link ranking | MRR |
| Peptides-func | 15,535 | 150.9 | 307.3 | No | graph | 10-task classif. | Avg. Precision |
| Peptides-struct | 15,535 | 150.9 | 307.3 | No | graph | 11-task regression | Mean Abs. Error |

**MNIST and CIFAR10** Dwivedi et al. (2023) (CC BY-SA 3.0 and MIT License) are derived from like-named image classification datasets by constructing an 8 nearest-neighbor graph of SLIC superpixels for each image. The 10-class classification tasks and standard dataset splits follow the

original image classification datasets, i.e., for MNIST 55K/5K/10K and for CIFAR10 45K/5K/10K train/validation/test graphs.

**PATTERN and CLUSTER** Dwivedi et al. (2023) (MIT License) are synthetic datasets sampled from Stochastic Block Model. Unlike other datasets, the prediction task here is an inductive node-level classification. In PATTERN the task is to recognize which nodes in a graph belong to one of 100 possible sub-graph patterns that were randomly generated with different SBM parameters than the rest of the graph. In CLUSTER, every graph is composed of 6 SBM-generated clusters, each drawn from the same distribution, with only a single node per cluster containing a unique cluster ID. The task is to infer which cluster ID each node belongs to.

**PascalVOC-SP and COCO-SP** Dwivedi et al. (2022b) (Custom license for Pascal VOC 2011 respecting Flickr terms of use, and CC BY 4.0 license) are derived by SLIC superpixelization of Pascal VOC and MS COCO image datasets. Both are node classification datasets, where each superpixel node belongs to a particular object class.

**PCQM-Contact** Dwivedi et al. (2022b) (CC BY 4.0) is derived from PCQM4Mv2 and respective 3D molecular structures. The task is a binary link prediction, identifying pairs of nodes that are considered to be in 3D contact (¡3.5Å) yet distant in the 2D graph (¿5 hops). The default evaluation ranking metric used is the Mean Reciprocal Rank (MRR).

**Peptides-func and Peptides-struct** Dwivedi et al. (2022b) (CC BY-NC 4.0) are both composed of atomic graphs of peptides retrieved from SATPdb. In Peptides-func the prediction is multi-label graph classification into 10 nonexclusive peptide functional classes. While for Peptides-struct the task is graph regression of 11 3D structural properties of the peptides.

### A.4.2 HYPERPARAMETERS

Table 5: Hyperparameters for five datasets from Long Range Graph Benchmark(LRGB)(Dwivedi et al., 2022b).

| Hyperparameter | PascalVOC-SP | COCO-SP | Peptides-func | Peptides-struct | PCQM-Contact |
|---|---|---|---|---|---|
| GPS Layers | 8 | 8 | 2 | 2 | 7 |
| Hidden dim | 68 | 68 | 235 | 235 | 64 |
| GPS-MPNN | GatedGCN | GatedGCN | GatedGCN | GatedGCN | GatedGCN |
| Heads | 4 | 4 | 4 | 4 | 4 |
| Dropout | 0.1 | 0.1 | 0.1 | 0.1 | 0.0 |
| Attention dropout | 0.5 | 0.5 | 0.5 | 0.5 | 0.5 |
| Graph pooling | – | – | mean | mean | – |
| Positional Encoding | LapPE-10 | – | LapPE-10 | LapPE-10 | LapPE-10 |
| PE dim | 16 | – | 16 | 16 | 16 |
| PE encoder | DeepSet | – | DeepSet | DeepSet | DeepSet |
| Batch size | 14 | 14 | 32 | 16 | 512 |
| Learning Rate | 0.001 | 0.001 | 0.001 | 0.001 | 0.0003 |
| Epochs | 200 | 150 | 250 | 250 | 200 |
| Warmup epochs | 10 | 10 | 5 | 5 | 10 |
| Weight decay | 0 | 0 | 0 | 0 | 0 |
| $\lambda^*$ | 1 | 1 | 1 | 1 | 1 |
| $\alpha$ | 0.1 | 0.1 | 0.1 | 0.1 | 0.1 |
| Parameters | 1,250,805 | 1,249,869 | 2,929,009 | 3,819,425 | 978,526 |

Table 6: Hyperparameters for four datasets from Dwivedi et al. (2023).

| Hyperparameter | **MNIST** | **CIFAR10** | **PATTERN** | **CLUSTER** |
|---|---|---|---|---|
| Layers | 5 | 5 | 4 | 20 |
| Hidden dim | 40 | 40 | 40 | 32 |
| MPNN | GatedGCN | GatedGCN | GatedGCN | GatedGCN |
| Heads | 4 | 4 | 4 | 8 |
| Dropout | 0.1 | 0.1 | 0 | 0.1 |
| Attention dropout | 0.1 | 0.1 | 0.5 | 0.5 |
| Graph pooling | mean | mean | – | – |
| Positional Encoding 0 | ESLapPE-8 | ESLapPE-8 | ESLapPE-10 | ESLapPE-10 |
| Batch size | 256 | 200 | 32 | 16 |
| Learning Rate | 0.001 | 0.001 | 0.0002 | 0.0002 |
| Epochs | 150 | 150 | 200 | 150 |
| Warmup epochs | 5 | 5 | 5 | 5 |
| Weight decay | 1e-5 | 1e-5 | 2e-5 | 1e-5 |
| $\lambda^*$ | 1 | 1 | 1 | 1 |
| $\alpha$ | 0.1 | 0.1 | 0.1 | 0.1 |
| Parameters | 275,465 | 275,545 | 222,213 | 1,211,330 |

