# OpenReview forum: "SFi-Former: Sparse Flow induced Attention for Graph Transformer"
_ICLR.cc/2025/Conference — ICLR 2025 Conference Withdrawn Submission_

### Official Review · Reviewer_HDsG · 2024-10-22

**Soundness:** 2
**Presentation:** 2
**Contribution:** 2
**Rating:** 3
**Confidence:** 5

**Summary:**

This paper develops a new attention mechanism for graph Transformers based on the energy function.  The authors provides the detailed theoretical analysis for the proposed method. The extensive results on diverse graph data mining tasks seem to show the promising performance of the proposed method.

**Strengths:**

1.  The authors provide the theoretical guarantee for the proposed method.
2.  The authors conduct the extensive experiments for performance comparison

**Weaknesses:**

1.  The paper is quite hard to read, the overall writing needs to be improved.
2.  The new self-attention mechanism seems to be independent with graphs.
3.  The experimental results are not convincing enough.

**Questions:**

1.  The calculation of the SFI-attention seems to be unrelated to the graph. It is more like a general attention mechanism for arbitrary input data. Can you provide more deep insights for modeling the graph structural data?
2.  The combination of adjacency matrix and attention matrix seems to be not reasonable in Eq. 11. The values in the adjacency matrix can be much larger than those in the attention matrix when the target node only has a few neighbors. The Fig.4 also proves the above claim. We can clearly observe that each node has larger attention values on themselves than those on other nodes. It is not suitable for information aggregation on graphs.
3.  The ablation study is insufficient. More ablation studies on other datasets are required. Moreover, the authors claim that all experiments are conducted based on the GraphGPS framework. Hence, I suggest the authors replace the original attention mechanism in GraphGPS with the proposed attention mechanism to demonstrate the effectiveness of the proposed method.
4.  The experimental results shown in Table 3 seem to indicate that the performance of SFi-Former is quite sensitive to the values of hyper-parameters. Moreover, Tab 5 and Tab 6 also suggest that SFi-Former needs a very very careful parameter tuning. The authors even carefully select the batch size, learning rate, epochs and warmup epochs to ensure the model performance, which makes the model comparison not convincing.  Hence, I believe the gain of SFI-Former is much from the parameter tuning, not the model designs.

---

### Official Review · Reviewer_VyYC · 2024-10-23

**Soundness:** 3
**Presentation:** 2
**Contribution:** 4
**Rating:** 5
**Confidence:** 4

**Summary:**

This paper introduces **SFi-Former**, a novel sparse flow-induced attention mechanism (SFi-attention) designed to address key limitations of dense attention in Graph Transformers (GTs), such as overfitting, over-globalization, and weak inductive bias. By leveraging energy optimization and an electric circuit analogy, the paper presents a new perspective on sparse attention, aiming to improve the generalization of Graph Neural Networks (GNNs). The experimental results demonstrate strong performance on several benchmark datasets, particularly in long-range dependency tasks.

**Strengths:**

- **Originality**: The paper presents a novel and theoretically rigorous approach to sparse attention in GNNs, which has the potential to significantly improve model generalization in long-range tasks.
- **Technical Contribution**: The use of energy-based optimization in the context of sparse attention is an innovative approach that adds depth to the understanding of attention mechanisms in GNNs.
- **Empirical Performance**: The paper shows strong results on several important datasets, which demonstrates the model’s potential in practical applications.

**Weaknesses:**

- **Clarity and Expression**: The paper’s presentation is hindered by complex and dense mathematical explanations, which may be difficult for readers to follow. Clearer explanations, diagrams, or more intuitive descriptions of the energy-based optimization and sparse attention would improve the paper’s accessibility.
- **Comprehensive Analysis**: The discussion of the model’s strengths and weaknesses is lacking. The paper should more thoroughly analyze the performance variations across different datasets and tasks, and provide insights into where the model excels and where it may fall short. This would provide a more balanced and holistic view of the contribution.
- **Conclusion and Future Directions**: The conclusion section is relatively brief and does not fully explore future research directions or practical applications. A more detailed discussion on the model’s potential use cases and limitations would strengthen the paper’s impact.

**Questions:**

- Could you provide more insights into the specific cases where the model struggles, and how you plan to address these issues in future work?
- What are the computational trade-offs of using energy-based optimization in SFi-attention compared to other sparse attention methods?
- How do you see this model being applied in real-world scenarios, and what further optimizations are needed for scalability?

---

### Official Review · Reviewer_KXrz · 2024-11-02

**Soundness:** 2
**Presentation:** 1
**Contribution:** 2
**Rating:** 3
**Confidence:** 4

**Summary:**

This paper proposes SFi-attention, a sparse attention mechanism designed to enhance Graph Transformers (GTs) for handling long-range dependencies in graph data. SFi-attention achieves sparsity by minimizing an energy function based on network flows with l1-norm regularization, aiming to mitigate overfitting, weak inductive bias, and over-globalizing effects.

**Strengths:**

1. The paper introduces a novel interpretation of graph transformers using circuit theory, offering a fresh perspective on model design.
2. The method is supported by a strong theoretical foundation.
3. The experiments shows the effectiveness of the proposed method.

**Weaknesses:**

1. The motivation is not sufficiently justified. While the abstract highlights the method’s design to address “weak inductive bias, overfitting, and over-globalizing problems,” there is little follow-up or comprehensive experimentation to demonstrate these effects.
2. The authors argue that their model learns sparser attention scores. But how the l1-norm can induce sparse attention is not well-explained. It seems that the sparsity is archived by a simple "hard" threshold. There is no experiment is to show the sparsity of the learned attention scores either.
3. In Section 3.1, the authors model the attention with circuit. Thus, it is very confusing in Section 3.2 that the author introduce friction. Thus I challenge the motivation and interpretation behind the proposed method.
4. In Section 3.3, how to compute the SFi-Attn $\boldsymbol{Z}^{*} (\boldsymbol{R}^h(\boldsymbol{X}), \boldsymbol{F}^h(\boldsymbol{X}))$ is not clear.
4. The ablation study is only conducted on three datasets. The authors are encouraged to show ablation studies on all the datasets. Meanwhile, the components show significant improvements on some dataset, while show little improvements on other datasets. The authors are encouraged to analyze this result. Beside, the sensitivity test on hyper-parameters is not enough.
5. There some typos in the paper. For example, in the equation of the attention in line 147 and line 148, the query and key should be swapped.

**Questions:**

1. Is the loss function in Equation (9) computed at every attention layer? What is the loss for the whole SFi transformer network?

---

### Official Review · Reviewer_LwVw · 2024-11-03

**Soundness:** 2
**Presentation:** 1
**Contribution:** 2
**Rating:** 5
**Confidence:** 3

**Summary:**

This paper introduces SFi-attention, a novel sparse attention mechanism for Graph Transformers (GTs) designed to process graph data with long-range dependencies. SFi-attention is noted to learn sparse pattern by minimizing an energy function based on network flows with l1-norm regularization.

**Strengths:**

1. This paper proposes a new interpretation of graph transformers from the perspective of circuit.
2. The theoretical foundation supporting the method adds depth and interest.

**Weaknesses:**

Weaknesses:
1. The motivation is not sufficiently justified. While the abstract highlights the method’s design to address “weak inductive bias, overfitting, and over-globalizing problems,” there is little follow-up or comprehensive experimentation to demonstrate these effects. Although a subsection in the experiments discusses reducing overfitting, the gap of 20%-30% remains quite large, and the authors should explain this discrepancy.
2. The authors assert that their model learns sparser attention scores, yet no experiments (such as statistical analysis or case studies) are provided to verify this claim.
3. Section 3.2 is difficult to follow. The introduction of friction in the context of electric energy is confusing, and the purpose of the “Soft” function design is unclear.
4. Section 3.3 also lacks clarity. The roles of Z, R(X), F(X), and Z(R(X),F(X)) in attention, as well as the computation of this attention, are not adequately explained.
5. In Table 2, the proposed method underperforms a baseline across most datasets. The authors should clarify the factors behind these results and specify conditions where their method performs better. Without such insight, the effectiveness of the proposed approach is questionable.
6. There some typos in the paper. For example, in the equation of the attention in line 147 and line 148, the query and key should be swapped.

**Questions:**

1. In Section 3.1, are resistors defined only on edges or between all node pairs? If defined between all pairs, how should one interpret a resistor between unlinked nodes?
2. Is the loss function in Equation (9) computed at every attention layer?

---

### Official Review · Reviewer_YzCj · 2024-11-04

**Soundness:** 3
**Presentation:** 3
**Contribution:** 2
**Rating:** 5
**Confidence:** 4

**Summary:**

This paper introduces SFi-Former, a graph-based model designed to capture long-range interactions using adjacency-enhanced and sparse attention mechanisms. It outperforms existing models on several benchmarks.

**Strengths:**

1. The paper addresses the challenge of capturing long-range interactions in graphs, which is crucial for modeling complex relationships in graph-based data.

2. The paper proposes the SFi-Former architecture, combining sparse attention mechanisms with adjacency enhancement. The theoretical analysis contributes to a deeper understanding of attention mechanisms with network flow optimization.

3. Extensive experiments on diverse datasets demonstrate the effectiveness of the proposed method.

**Weaknesses:**

1. The paper suggests limited prior research on energy-based models in graph transformers; however, existing work such as DIFFormer [1] and SGFormer [2] has explored similar energy-constrained frameworks.

2. While the paper presents a theoretical explanation of its sparse attention mechanism, it omits consideration of the adjacency matrix and message-passing components in this framework. Many recent theoretical studies [1, 2, 3, 4] on GNNs and GTs employ energy-function-based models. Linking the proposed framework with established theoretical insights in this area would enhance the theoretical contribution.

3. An additional ablation test excluding the friction parameter, while maintaining sparse constraints on Z, would clarify its specific role. Additionally, the paper could benefit from further discussion on issues of weak inductive bias, overfitting, and over-globalizing within the experimental results.

4. The paper suggests that its sparse attention can replace attention modules in existing graph transformers. However, comparisons of model performance before and after substituting with sparse attention are currently not provided. It would also be beneficial to test if simple sparsity regularization techniques (e.g., L1, L2, or entropy regularization) on attention coefficients in existing models achieve similar gains.

5. The performance gains in the second experiment are minimal and within random variation, which may raise questions regarding the method’s effectiveness on GNN benchmark datasets.

6. Although the authors acknowledge that addressing GTs' computational bottleneck is beyond the scope, a brief analysis of the model's complexity and computational cost would be useful. This analysis would help assess the trade-offs between performance improvements and the associated computational overhead.


[1] Difformer: Scalable (Graph) Transformers Induced by Energy Constrained Diffusion, ICLR 2023.

[2] Simplifying and Empowering Transformers for Large-Graph Representations, NeurIPS 2023.

[3] Elastic Graph Neural Networks, ICML 2021.

[4] Interpreting and Unifying Graph Neural Networks with An Optimization Framework, WWW 2021.

**Questions:**

1. Could the authors clarify how this work differs from and relates to existing research, such as DIFFormer and SGFormer?

2. Could the authors perform an ablation test without the friction parameter to clarify its role?

3. Could the authors provide performance comparisons with and without sparse attention in existing GTs and explore if applying sparsity regularization on attention coefficients yields similar gains?

4. Given minimal improvements in the second experiment, could the authors explain the method’s effectiveness on GNN benchmarks?

5. While computational bottlenecks are not the focus, would the authors provide a brief analysis of model complexity to assess scalability?

---

### Note · Authors · 2024-11-18

I have read and agree with the venue's withdrawal policy on behalf of myself and my co-authors.